# Adherence to Covid-19 mitigation measures and its associated factors among health care workers at referral hospitals in Amhara regional state of Ethiopia

**Agazhe Aemro**[1], **Beletech Fentie**[2]*, **Mulugeta Wassie**[1]

1 Department of Medical Nursing, College of Medicine and Health Sciences, University of Gondar, Gondar, Ethiopia, 2 Department Pediatrics and Child Health Nursing, College of Medicine and Health Sciences, University of Gondar, Gondar, Ethiopia

* beletechfentie@gmail.com

## Abstract

### Introduction

With fragile health care systems, sub-Saharan Africa countries like Ethiopia are facing a complex epidemic, and become difficult to control the noble coronavirus. The use of COVID-19 preventive measures is strongly recommended. This study aimed to assess the adherence of COVID-19 mitigation measures and associated factors among health care workers.

### Methods

A facility-based cross-sectional study was conducted among health care workers at referral hospitals in the Amhara regional state of Ethiopia from May 15 to June 10; 2021. It was a web-based study using an online questionnaire. STATA 14.2 was used for data analysis. Variables with a p-value<0.05 at 95% confidence level in multivariable analysis were declared as statistically significant using binary logistic regression.

### Result

Adherence to COVID-19 mitigation measures was 50.24% in the current study. The odd of adherence of participants with a monthly income of ≥12801birr was 15% whereas the odds of adherence of participants who hesitate to take the COVID 19 vaccine were 10% as compared to those who don't hesitate. Participants who had undergone COVID-19 tests adhered 6.64 times more than their counterparts. Those who believe adequate measurements are taken by the government adhered 4.6 times more than those who believe not adequate. Participants who believe as no risk of severe disease adhered 16% compared to those with fear of severe disease. Presence of households aged >60years adhered about 7.9 times more than with no households aged>60. Participants suspected of COVID-19 diagnosis adhered 5.7 times more than those not suspected.

**Data Availability Statement:** All relevant data are within the manuscript and its Supporting information files.

**Funding:** The author(s) received no specific funding for this work.

**Competing interests:** The authors have declared that no competing interests exist.

## Conclusion

In this study, a significant proportion of healthcare workers did not adhere to COVID-19 mitigation measures. Hence, giving special attention to healthcare workers with a monthly income of ≥12801 birr, being hesitant towards COVID-19 vaccine, being aged 26–30, and perceiving no risk of developing a severe infection is crucial to reduce non-adherence.

## Introduction

The pandemic of COVID-19 entered Africa continent by the termination of February 2020 afterward it was professed a public health emergency of Worldwide Concern by the world health organization [1]. With fragile health care systems, African countries like Ethiopia are facing a complex COVID-19 epidemic, and it becomes a unbreakable duty to switch the virus reservoir, from where the virus may be introduce again to other regions [2].

Globally, COVID-19 affected more than 119.7 million people and 2.6 million deaths occurred [3] whereas in Africa over 4 million cases and 107 thousand deaths have been confirmed [4]. Considering the pandemic and lack of efficient management, government regulators' in the globe designed different mitigation methods to battle the spread of the pandemic [5, 6]. To control the pandemic transmission, world health organization endorses reducing contact, early identification and isolation of cases, personal and material hygiene measures [6, 7].

As part of these measures, the use of face masks, hand washing, physical distancing, cough etiquette, and avoidance of crowded places are strongly recommended [7]. Even though adherence to preventive measures is the only means to tackle the disease, reluctance to do so has been reported to be a major problem everywhere [8].

Health care professionals are facing more workload, mental distress, scarcity of quality personal protective equipment, social exclusion, absence of motivations, coordination and good leadership throughout their service [9].

The good adherence to the COVID-19 pandemic mitigation measures was 51.04% and 8.3% in different Ethiopian studies conducted in the general community, but there is no information among health care providers [5, 8].

A substantial number of health care workers were reported to be infected with COVID-19 within the first six months of the COVID-19 pandemic, with the occurrence of hospitalization of 15.1% and mortality of 1.5% [10].

Health care providers are also facing many challenges like physical and mental affects, stigma and discrimination, fear of infection, and overall trying their best to keep it together. Health care providers could forget the mitigation measures of COVID-19 due to high workload, stress and related factors which will cause significant disruption of prevention chains of the disease [11, 12].

As far our search, there is no research conducted among health care workers in the current study setting and it is also true in the country as large. Therefore, this study intended to assess the adherence of COVID-19 mitigation measures and their associated factors among health care providers in the Amhara region regional state of Ethiopia.

## Methods and materials

### Study design, period and setting

A facility-based cross-sectional study was conducted from May 15 to June 10; 2021. It was a Web-based anonymous study using an online questionnaire. The study was conducted at

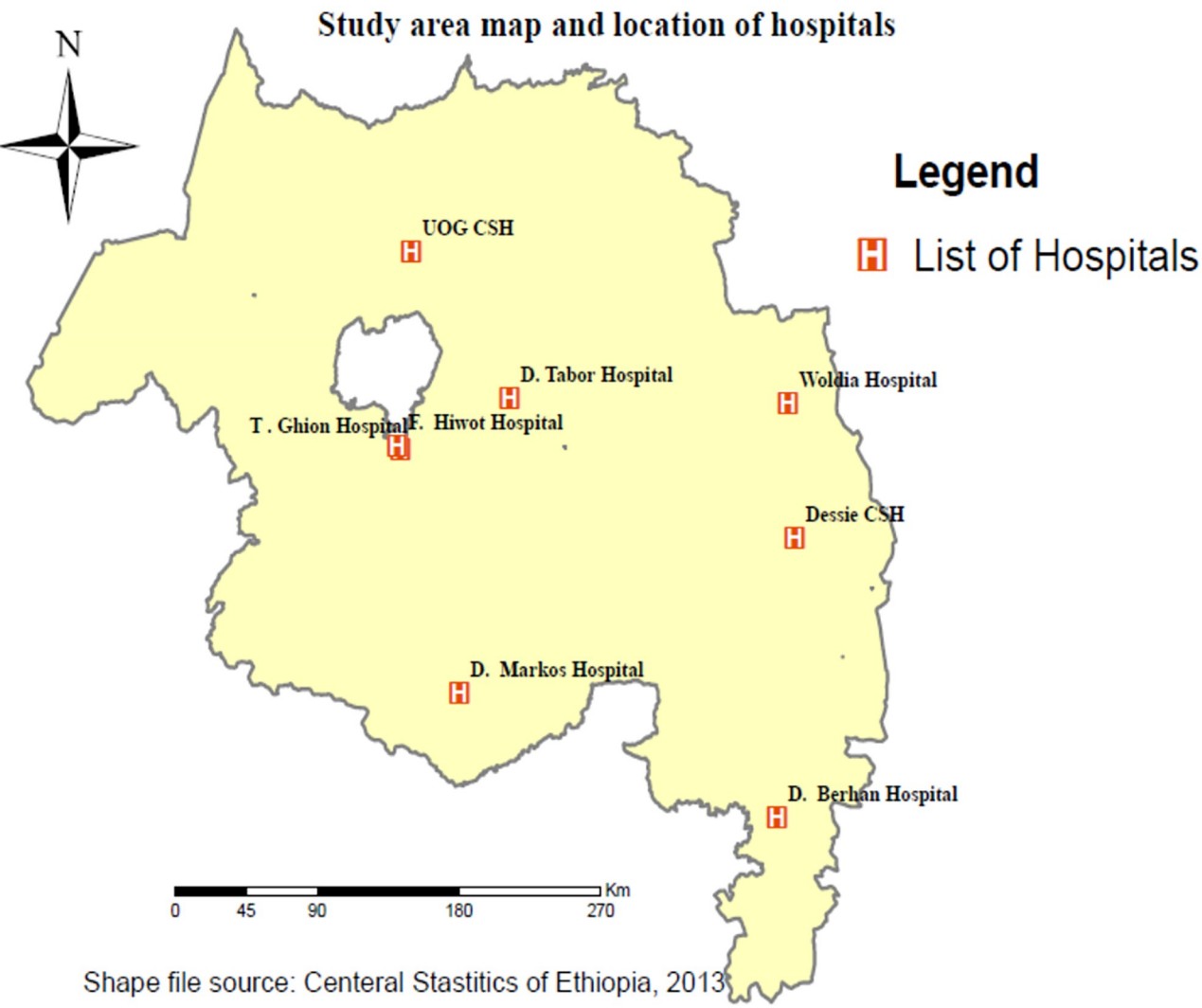

**Fig 1. Schematic presentation of referral hospitals in Amhara regional state, Ethiopia.**

referral hospitals in the Amhara regional state. According to the Amhara national regional health bureau annual performance report, the region has 81 hospitals, 858 health centers, and 3560 health posts. Among the hospitals in the region, the University of Gondar, Dessie, Felege-Hiwot, Tibebe-Ghion, Debre-Markos, Waldiya, Debre Tabor, and Debre Berhan are referral hospitals (Fig 1). The health care professionals working in these hospitals are estimated to be 4,000 [13, 14].

## Study participants

Telegram and email (the most popular social media platforms in Ethiopia) were used to promote and circulate the survey link to the participants. Data collectors in each hospital were asked to distribute the survey link to the randomly selected contacts in each hospital. The participants were informed that their participation was based on voluntariness, and consent was implied through their completion of the questionnaire. The respondents working during the data collection period were included in the current study.

## Sample size determination

The sample size was determined using the single population proportion formula taking the proportion of compliance to the COVID19 preventive measures 22% [15], 95% confidence interval, and 4% marginal error. After adding a 5% non-response rate, the final sample size was ∼ 433.

## Sampling procedure

There are about 4,000 health care workers in Amhara regional state referral hospitals(906 in Gondar hospital, 320 in Debre tabor hospital,255 in Tibebe Ghione hospital,917 in Felege Hiot hospital,430 in Debre Markos Hospital,604 in Dessie hospital,300 in Waldiya hospital and 270 in Debre Berhan hospital). The entire sample size was first allocated proportionally to those eight referral hospitals. In order to select study participants from each hospital, first, the list of active healthcare workers during the study period was taken from the human resource management office of each hospital. Since the data was collected using telegram or e-mail, healthcare workers with no recorded information at either of these two addresses were excluded from the study. After that, a random number was generated on the computer, and by using this number and based on the allocated sample size, study participants were selected. Finally, the link of the questionnaire was given to the data collectors and forwarded to randomly select health care workers of respected hospitals, using e-mail or telegram. The link was forwarded to each hospital's data collector to avoid coverage bias and to be representative.

## Operational definitions

**Good adherence of COVID-19 mitigation measures.** Adherence in the current study was measured as participants who adhered (responded "yes") to all of the three basic preventive measures (Wearing a mask, keeping physical distancing of a minimum of 2 meters, and Handwashing a minimum of ≥6 times/ day) and measured 'Yes' or 'No' answers to the questions. The specific questions used to assess the adherence of mask wearing, hand washing and physical distancing were asked as "have you wear face mask every time you leave home and never remove it from the face? (Yes/ No), do you wash your hand with soap at least six times per day during the Covid-19 pandemic? (yes/no) and are you fully compliance with physical distancing (≥2 meter) during the Covid-19 pandemic?(Yes/no) respectively. Individual participants who respond "Yes" for each component were adhered for mitigation measures in the current study. Participants who did not adhere even one of the three components of the mitigation measures were considered not adhered at the whole. We have summed all the three components and calculated the whole adherence.

**Health care worker (HCW).** Any member of the health care unit that includes medical doctors, pharmacists, physiotherapists, midwifery, laboratory technologists, nursing professionals, or any other person in the course of his or her professional activities who may prescribe, administer, or dispense a medicinal product to an end-user [16].

**Vaccine hesitancy.** World Health Organization (WHO) declared vaccine hesitancy as "the reluctance or refusal to vaccinate despite the availability of vaccines" [17]. Respondents said to be hesitant to the vaccine if they respond "No" to the question "By the time you get a chance for Covid-19 vaccine, will you take the vaccine without any refusal?".

**Perceived susceptibility COVID-19 infection.** Refers to a participant's subjective perception of the risk of acquiring COVID-19 and is measured as High, Moderate, Low, No risk, or not sure [18].

**Perceived severity of COVID-19 infection.** Refers to a person's subjective perception of the seriousness of contracting COVID-19 and measured as High, Moderate, Low, No risk, or not sure [18].

## Data processing and analysis

The responses from the participants were downloaded in Excel using Google Forms. The data were checked for completeness and consistency, then compiled and coded. Then, it was exported to STATA version 14.2 statistical software for analysis. A binary logistic regression was employed to identify factors associated with adherence to COVID-19 mitigation measures. Initially, bivariate analysis was done, and variables with a p-value of 0.2 or below were identified as candidates for multivariable analysis. Then, multivariable analysis was done, and the adjusted odd ratio with a 95% confidence interval was computed and interpreted. A p-value of less than 0.05 is the cut-off point for determining the significance of an association. Finally, the result of the study was presented in text and tables.

## Data quality assurance

The web-based self-administered questionnaire was pretested by taking 5% of the sample size before the actual data collection period. Afterward, the pretests, amendments to the tool, like formatting were corrected. The tool was first developed in the English language and was translated into the local language (Amharic) with back translation to English to check its consistency. Moreover, Cronbach's alpha value was calculated to check the tools' reliability and the value of an item score was 0.892.

## Ethics approval and consent to participate

This study was approved by the institutional review board (IRB) of the University of Gondar. Written informed consent was obtained from each participant using communication channels (telegram and email) and those who agreed to participate were included in the study. Respondents were informed that their participation was voluntary and their confidentiality was maintained by avoiding registration of personal identifiers like names on the questionnaire and also, no raw data was given to anyone other than the investigator. In addition, the raw data is secured by a strong computer password.

# Results

## Socio-demographic characteristics of study participants

From the total 433 samples, 418 participants completed the questionnaire that yielded a 96.5% response rate. The mean age of study participants was 29.95 in the current study. More than half of the participants were under the age category of 26–30 years and nearly two-thirds were males. About 54% were married, 55% BSc and below educational level. The majority of the study participants have a monthly income in the category of 6991–12800 birr. Based on family size, 53.35% have less than or equal to 2 and nearly one thirds (31.58%) have children with school-age (Table 1).

## COVID-19 related characteristics of study participants

Nearly two-thirds (63.64%) of the participants were socially isolated because of their profession. About 59% underwent the COVID test and 44% were confident in health care services delivered on their institution whereas 45.69% got unclear information by health authorities related to the COVID-19 pandemic. Only100 (23.92%) believe measurements taken by the

**Table 1. Socio-demographic characteristics of study participants (N = 418).**

| Variables | Category | Frequency | Percent (%) |
|---|---|---|---|
| Age | ≤25 | 49 | 11.72 |
| | 26–30 | 234 | 55.98 |
| | ≥31 | 135 | 32.30 |
| Sex | Female | 129 | 30.86 |
| | Male | 289 | 69.14 |
| Marital status | Single | 194 | 46.41 |
| | Married | 224 | 53.59 |
| Educational status | BSc and below | 230 | 55.02 |
| | MSc and above | 188 | 44.98 |
| Monthly income | <6990 | 27 | 6.46 |
| | 6991–12800 | 357 | 85.41 |
| | ≥12801 | 34 | 8.13 |
| Family size | ≤2 | 223 | 53.35 |
| | 3–4 | 128 | 30.62 |
| | ≥5 | 67 | 16.03 |
| School-age children | No | 286 | 68.42 |
| | Yes | 132 | 31.58 |

national government related to COVID-19 preventive measures are adequate. More than half (52.39%) of participants reported that they are at higher risk of COVID-19 infection but 53.35% believe they are at low risk to develop the severe disease if infected with the coronavirus. Nearly two-thirds (63.64%) had good compliance on social isolation if suspected to COVID-19 whereas 57.89% were suspected of COVID-19 diagnosis. About 55% perceive that their health status was very good. Only 3.83% have autoimmune diseases taking steroidal drugs. Nearly 54% of the participants were willing to take the COVID-19 vaccine but about 19% are confident in the current vaccine (Table 2).

## Adherence towards COVID-19 mitigation measures

The Adherence towards COVID-19 mitigation measures among health care workers in the current study was 50.24[95%CI (45.44–55.04)]. Adherence to COVID-19 measures was 71.29%, 73.21%, and 56.94% for wearing a mask, washing hands ≥6 times per day based on WHO hand washing rules, and physical distancing of at least 2 meters respectively (Fig 2).

## Factors associated with adherence of COVID-19 mitigation measures

Binary logistic regression was employed to identify independent factors that can affect the outcome variable. In bivariable analysis, monthly income, hesitancy to take COVID-19 vaccine, age, marital status, undergone COVID-19 test, the information given by health authority, measures taken by the national government, the risk to get COVID-19 disease, risk of severe COVID-19 disease, household age >60 years, suspected to COVID-19 infection, Comorbidity and confident on the current COVID-19 vaccine were associated with the outcome variable.

But in multivariable analysis, monthly income, hesitancy to take COVID-19 vaccine, age, undergone COVID-19 test, measures taken by the national government, household aged >60 years, and suspected to COVID-19 infection were statistically significant variables that affected adherence of COVID-19 mitigation measures.

**Table 2. COVID-19 related characteristics of the study participants (N = 418).**

| | | | |
|---|---|---|---|
| Social isolation | No | 152 | 36.36 |
| | Yes | 266 | 63.64 |
| Undergone COVID test | No | 172 | 41.15 |
| | Yes | 246 | 58.85 |
| Confident in health care services | Not confident | 194 | 46.41 |
| | Confident | 184 | 44.02 |
| | Very confident | 40 | 9.57 |
| Information by health authorities | Clear | 134 | 32.06 |
| | Inconsistent | 93 | 22.25 |
| | Unclear | 191 | 45.69 |
| Measurements by Gov't | Not very adequate | 186 | 44.50 |
| | Not adequate | 132 | 31.58 |
| | Adequate | 100 | 23.92 |
| Risk to get COVID-19 infection | Low | 96 | 22.97 |
| | Moderate | 103 | 24.64 |
| | High | 219 | 52.39 |
| Risk to sever COVID-19 disease | Moderate/high | 115 | 27.51 |
| | Low | 223 | 53.35 |
| | No/not sure | 80 | 19.14 |
| Households age >60 years | No | 355 | 84.93 |
| | Yes | 63 | 15.07 |
| Compliance to social isolation | No | 152 | 36.36 |
| | Yes | 266 | 63.64 |
| Suspected COVID-19 Diagnosis | No | 176 | 42.11 |
| | Yes | 242 | 57.89 |
| Undergone COVID-19 test | No | 172 | 41.15 |
| | Yes | 246 | 58.85 |
| Perception of your health status | Good | 183 | 43.78 |
| | Very bad | 7 | 1.67 |
| | Very good | 228 | 54.55 |
| Autoimmune problem /taking steroid | No | 402 | 96.17 |
| | Yes | 16 | 3.83 |
| Will you take COVID-19 vaccine | No | 192 | 45.93 |
| | Yes | 226 | 54.07 |
| Confident in the current COVID-19 vaccine | Confident | 80 | 19.14 |
| | Not confident | 100 | 23.92 |
| | Not very confident | 230 | 55.02 |
| | Very confident | 8 | 1.91 |

Study participants with a monthly income of ≥12801birr adhered to COVID-19 measures 15% taking monthly income of ≤6990 as reference [AOR = 0.15, 95%CI (0.02–0.92)]. Participants who hesitate to take COVID 19 vaccine adhered 10% [AOR = 0.10, 95%CI (0.04–0.25)] as compared to those who don't hesitate. Those participants with the age group of 26–30 years adhered to mitigation measures 9% [AOR = 0.09, 95%CI (0.02–0.39)] compared to age groups <26 years. Study participants who underwent the COVID-19 test adhered to about 6.6 [AOR = 6.64, 95%CI (3.10–14.22)] times more than those who didn't undergo the test. Participants who believe adequate measurements are taken by the government adhered to about 4.6

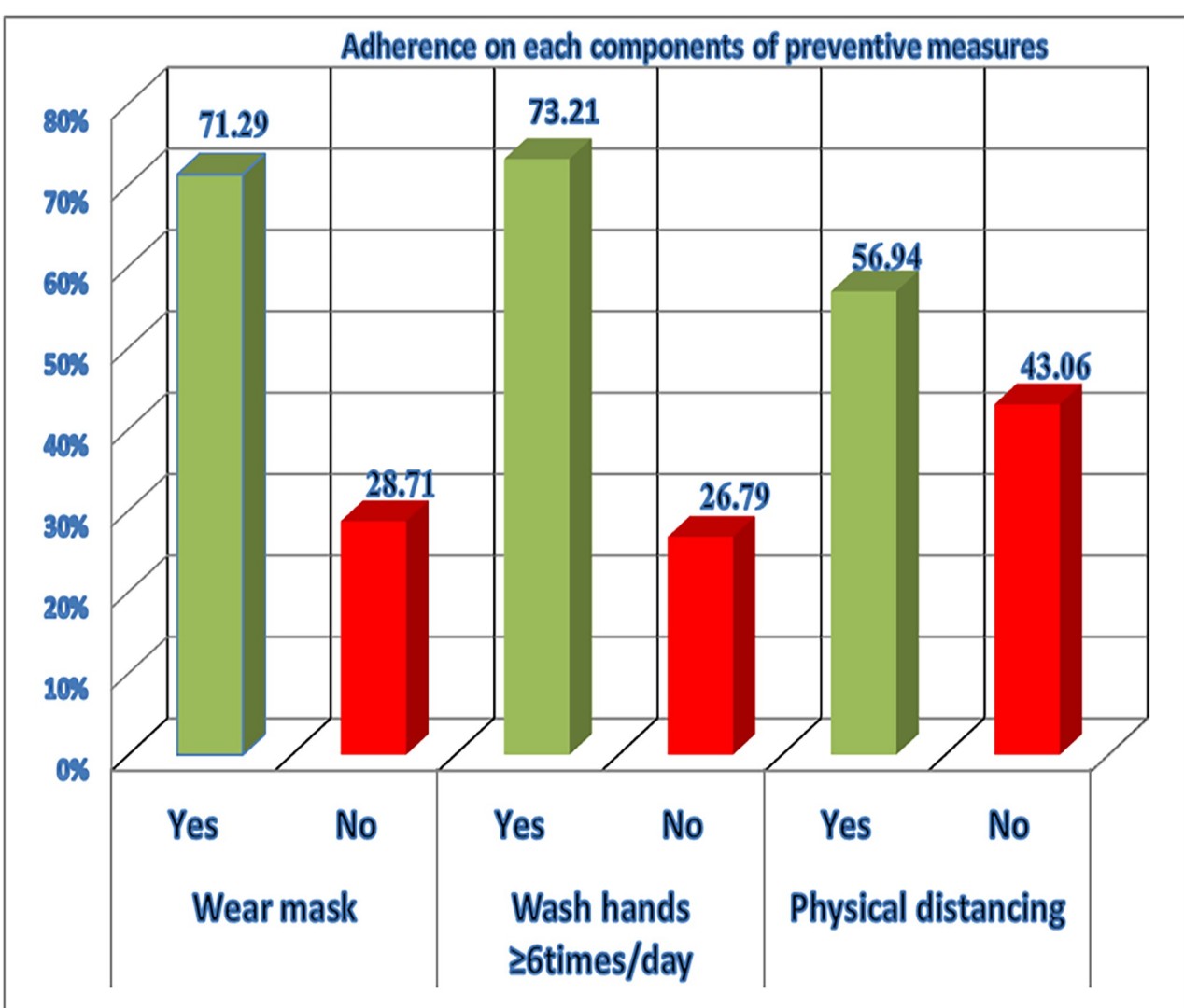

**Fig 2. Distribution of adherence of COVID-19 mitigation measures among health care workers in referral hospitals of Amhara regional state of Ethiopia.**

[AOR = 4.60, 95%CI (1.66–12.78)] times more than those who believe measurements are not adequate. Participants who believe with no risk of severe COVID-19 disease adhered 16% [AOR = 0.16, 95%CI (0.06-.46)] as compared to those with fear of severe COVID-19 disease. Participants who have households aged >60 years adhered about 7.9[AOR = 7.94, 95%CI (3.14–20.04)] times more than those with no households aged>60 years and those participants suspected to COVID-19 infection adhered to mitigation measures about 5.7 [AOR = 5.74, 95% CI (1.81–18.16)] times more than those who didn't suspect (Table 3).

## Discussion

The current study aimed to determine adherence to COVID-19 mitigation measures and their associated factors. The adherence to COVID-19 mitigation measures among the participants was found to be 50.24%. The highest adherence (73.21%) was reported for handwashing whereas the lowest (56.94%) was reported for physical distancing.

**Table 3. Factors associated with adherence of COVID-19 mitigation measures among health care workers in referral hospitals of Amhara regional state of Ethiopia (N = 418).**

| Variables | Category | COR | AOR | P-value | 95% CI |
|---|---|---|---|---|---|
| Monthly income | ≤6990 | 1 | 1 | 1 | |
| | 6991–12800 | 0.21* | 0.29 | 0.105 | (0.06–1.29) |
| | ≥12801 | 0.18* | 0.15 | **0.041** | (0.02–0.92) |
| Hesitancy to COVID 19 vaccine | No | 1 | 1 | 1 | |
| | Yes | 0.09* | 0.10 | **<0.001** | (0.04–0.25) |
| Age | <26 | 1 | 1 | 1 | |
| | 26–30 | 0.53* | 0.09 | **0.001** | (0.02–0.39) |
| | ≥31 | 0.97 | 0.25 | 0.065 | (0.05–1.08) |
| Marital status | single | 1 | 1 | 1 | |
| | Married | 0.72* | 1.33 | 0.422 | (0.66–2.68) |
| Undergone COVID-19 19 test | Yes | 8.28* | 6.64 | **<0.001** | (3.10–14.22) |
| | No | 1 | 1 | 1 | |
| Information health by authorities | clear | 0.89 | 1.79 | 0.250 | (0.66–4.88) |
| | Inconsistent | 0.37* | 0.41 | 0.071 | (0.15–1.07) |
| | Unclear | 1 | 1 | 1 | |
| Measures by Gov't | Not very adequate | 1 | 1 | 1 | |
| | Not adequate | 0.29* | 0.66 | 0.299 | (0.29–1.45) |
| | Adequate | 1.28 | 4.60 | **0.003** | (1.66–12.78) |
| Risk to get COVID-19 disease | low | 1 | 1 | 1 | |
| | Moderate | 0.89 | 2.40 | 0.136 | (0.75–7.61) |
| | High | 3.09* | 2.13 | 0.125 | (0.81–5.60) |
| Fear to risk of sever COVID-19 disease | No risk | 0.13* | 0.16 | **0.001** | (0.06-.46) |
| | Low risk | 0.57* | 0.45 | 10.07 | (0.18–1.08) |
| | Moderate/high risk | 1 | 1 | 1 | |
| Household with age >60yrs | Yes | 1.8 0* | 7.94 | **<0.001** | (3.14–20.04) |
| | No | 1 | 1 | 1 | |
| Suspected to COVID-19 infection | Yes | 12.51* | 5.74 | **0.003** | (1.81–18.16) |
| | No | 1 | 1 | 1 | |
| Comorbidity | No | 1 | 1 | 1 | |
| | Yes | 0.14 | 0.35 | 0.347 | (0.04–3.08) |
| Confident on COVID-19 vaccine | Not confident | 0.50 | 0.78 | 0.618 | (0.29–2.07) |
| | Confident | 1 | 1 | 1 | |

* = variables associated with the outcome variable at p-value<0.2, 1 = reference category of the respected variable.

The current finding of adherence was lower than the study conducted in Saudi Arabia (82%), the United Kingdom(80%), and the Kingdom of Saudi Arabia(80.9%) [19–21]. The possible reasons for this difference might be the countries' policy to prevent the pandemic, the monthly income difference of the study participants which might affect buying abilities of face masks, the data collection period differences in which all the studies conducted before the current study when vaccines were not found. But the current finding is more than the studies conducted in Western Ethiopia (22%) and southeast Ethiopia (21.6%) [15, 22]. The possible justification of the differences of the findings might be differences in COVID-19 prevention policies of the respected health institutions in the specified regions even though they are found in the same country.

Different independent variables in the current study affected the outcome variable. Monthly income, vaccine hesitancy, age, undergone COVID19 test, measurements taken by the national government, perception of the severity of the disease, presence of households with age>60 years and suspected to COVID-19 diagnosis significantly affected the adherence of COVID-19 measures in different directions.

Study participants in the current study with a monthly income of ≥12801birr adhered to COVID-19 measures less than those with a monthly income of ≤6990 birrs. This might be participants with low monthly income could use public transportation which might increase fear to acquire COVID-19 infection and cause them to adhere more [23, 24].

Participants who hesitate to take COVID -19 vaccines adhered lower than those who are volunteers to take the vaccine. The current study finding is supported by different studies conducted in Germany and China [25, 26]. The possible reason could be those who hesitate to take the vaccine might be individuals who believe COVID -19 is not a severe disease and even there is no such disease [27–29].

Study participants with age groups of 26–30 years adhered to mitigation measures lower than those with age groups of <26 years. The current finding is in contradiction with the study conducted in South Ethiopia among the general community [30]. The discrepancy might be due to the current study conducted among health care workers but the previous study was conducted among the general community. The possible justification for the current study would be younger professionals might abide by mitigation measures more than elders due to negligence [31].

The experienced COVID-19 test increased the participants' adherence in the current study. This might be as the participants who believe the existence of the pandemic is high and resulted to undergo COVID-19 test and consequently adhered to the mitigation measures than those who didn't experience the COVID-19 test [1, 32]. Similarly, study participants who think that adequate measurements are taken by the national government adhered to mitigation measures more than those who think not taking adequate measurements. This could be those thinking the national government is taking adequate measurement trusted the national policies related to the pandemic and consequently adhered more [33, 34].

Study participants who perceived the severity of the disease as high adhered more than those who perceived no risk. Naturally, those who perceive the disease as severe are more committed to prevent it [35]. Participants who have households aged>60 years adhered more than those with no. The current finding is in line with the study conducted in Slovenia [36] This might be because individuals with age >60 years are at the risk of getting severe complications of the COVID -19 like death [37]. Therefore, those participants with households of age >60 years adhered more to prevent such complications of their households.

Another factor that increased the adherence to theCOVID-19 mitigation measures was suspected to COVID-19 infection. This finding is in agreement with the study finding conducted in Congo [38]. This might be as those suspected of the disease would not be allowed to enter the working area and consequently adhere to the preventive measures [39].

## Conclusion

This study found lower adherence to COVID-19 mitigation measures among health care workers. Greater monthly income, hesitate to take the vaccine and older age decreased the adherence whereas undergone COVID-19 test, adequate measurement by the government, believing severity of the disease, households with age >60 years and suspected to COVID-19 diagnosis increased the adherence of mitigation measures. It is better to boost the practice of

health care workers on the prevention methods of the COVID-19 pandemic in the current study setting since the adherence of the mitigation measures is lower than the recommended.

## Limitations of the study

Since this study is cross-sectional, it shares the limitations of a cross-sectional study design. Social desirability bias could be introduced through study participants since the data collection technique was self-administered. To avoid the mentioned bias, the authors recommend doing further investigation using observational checklists.

## Supporting information

**S1 Dataset.**
(XLS)

**S1 File.**
(DOCX)

## Acknowledgments

The authors would like to acknowledge the healthcare workers for their collaboration during the data collection. Our gratitude also goes to data collectors in each hospital. Last but not least, we would like to pass our thanks to the University of Gondar for providing ethical clearance to conduct this study.

## Author Contributions

**Conceptualization:** Agazhe Aemro, Beletech Fentie, Mulugeta Wassie.

**Data curation:** Agazhe Aemro.

**Formal analysis:** Agazhe Aemro, Beletech Fentie, Mulugeta Wassie.

**Funding acquisition:** Agazhe Aemro, Beletech Fentie, Mulugeta Wassie.

**Investigation:** Agazhe Aemro, Beletech Fentie, Mulugeta Wassie.

**Methodology:** Agazhe Aemro, Beletech Fentie, Mulugeta Wassie.

**Project administration:** Agazhe Aemro, Beletech Fentie, Mulugeta Wassie.

**Resources:** Agazhe Aemro, Beletech Fentie, Mulugeta Wassie.

**Software:** Agazhe Aemro, Beletech Fentie, Mulugeta Wassie.

**Supervision:** Agazhe Aemro, Beletech Fentie, Mulugeta Wassie.

**Validation:** Mulugeta Wassie.

**Visualization:** Agazhe Aemro, Beletech Fentie, Mulugeta Wassie.

**Writing – original draft:** Beletech Fentie, Mulugeta Wassie.

**Writing – review & editing:** Agazhe Aemro, Beletech Fentie, Mulugeta Wassie.

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
