## [Decision Letter · Decision Letter 0]

21 Jan 2022

PONE-D-21-38230Adherence of Covid-19 mitigation measures and its associated factors among health care workers at Referral Hospitals in Amhara Regional state of Ethiopia.PLOS ONE

Dear Dr. Beletech Fenti

Thank you for submitting your manuscript to PLOS ONE. After careful consideration, we feel that it has merit but does not fully meet PLOS ONE’s publication criteria as it currently stands. Therefore, we invite you to submit a revised version of the manuscript that addresses the points raised during the review process.

We look forward to receiving your revised manuscript.

Kind regards,

Paavani Atluri

Academic Editor

PLOS ONE

Journal Requirements:

● A clean copy of the edited manuscript (uploaded as the new *manuscript* file).

5. PLOS requires an ORCID iD for the corresponding author in Editorial Manager on papers submitted after December 6th, 2016. Please ensure that you have an ORCID iD and that it is validated in Editorial Manager. To do this, go to ‘Update my Information’ (in the upper left-hand corner of the main menu), and click on the Fetch/Validate link next to the ORCID field. This will take you to the ORCID site and allow you to create a new iD or authenticate a pre-existing iD in Editorial Manager. Please see the following video for instructions on linking an ORCID iD to your Editorial Manager account: https://www.youtube.com/watch?v=_xcclfuvtxQ.

7. Thank you for submitting the above manuscript to PLOS ONE. During our internal evaluation of the manuscript, we found significant text overlap between your submission and the following previously published works, some of which you are an author.

- https://journals.plos.org/plosone/article?id=10.1371%2Fjournal.pone.0257373

Please revise the manuscript to rephrase the duplicated text, cite your sources, and provide details as to how the current manuscript advances on previous work. Please note that further consideration is dependent on the submission of a manuscript that addresses these concerns about the overlap in text with published work.

Reviewers' comments:

Reviewer's Responses to Questions

**Comments to the Author**

1. Is the manuscript technically sound, and do the data support the conclusions?

Reviewer #1: Yes

Reviewer #2: Yes

Reviewer #3: Yes

Reviewer #4: Yes

Reviewer #5: Partly

2. Has the statistical analysis been performed appropriately and rigorously? 

Reviewer #1: Yes

Reviewer #2: Yes

Reviewer #3: Yes

Reviewer #4: Yes

Reviewer #5: N/A

3. Have the authors made all data underlying the findings in their manuscript fully available?

Reviewer #1: Yes

Reviewer #2: Yes

Reviewer #3: Yes

Reviewer #4: Yes

Reviewer #5: No

4. Is the manuscript presented in an intelligible fashion and written in standard English?

Reviewer #1: Yes

Reviewer #2: Yes

Reviewer #3: Yes

Reviewer #4: Yes

Reviewer #5: Yes

5. Review Comments to the Author

Reviewer #1: 1. since your study is a form of survey study, could you add study map?? to have more description for the reader?

2. correct place for ethics approval and consent to participate subtitle, and no need to write duplicate subtitle

3. No need to describe the study area / objective under discussion again!!

4. duplicate statements under abstract and conclusion

Reviewer #2: When you are writing the statement of the problem, it was nice if you put the paragraphs as follows;

• Concise description of the problem (severity, which group is affected, the distribution of the problem, what contribute to the problem, the consequences of the problem. what policies, and strategies are in palace to combat the problem , what is known, what is not known, why you are interested in the topic ( what gaps exist)

• You can use few studies to describe the problem but it’s recommended to summaries studies with similar findings in one statement

Better to put in such way Binary logistic regression was be employed to identify factors associated with adherence of COVID-19 mitigation measures. Initially bivariate analysis was done and variables with p-value of below 0.2 was identified as candidate for multi-variable analysis. Then multi-variable analysis was done and adjusted odd ration was computed and interpreted. A p-value less than 0.05 is cut-off point for determining the significance of association. Result of the study was presented in text, table and graphs.

Reviewer #3: None

Reviewer #4: I appreciate the authors for doing a research on the current pandemic disease. But I have some concerns.

1. Abstract is OK.

2. Methods: Please provide further detail how Random selection was carried out to select study participants.

3. Please try to provide the detail of the specific questions used to assess adherence level for all the three components.

4. Discussion: The justification provided by the authors on the discrepancy between the studies conducted in other part of the same country may be due to the difference in tool used to assess adherence. Please provide a clear justification why you preferred using a tool with only three components to assess adherence level when others used different tool( discussion part second paragraph line 13-14)

5. In discussion, the author only compared their finding with other studies on the adherence level and no comparisons were made with other studies for factors affecting adherence level.

6. Discussion part should cover the interpretation of the finding, comparison with other studies, explanation for discrepancies if it exists, and the limitation of the study. The discussion part could be more than what the authors provided.

Reviewer #5: dear authors thank you for your effort, i have some comments and questions in your work

Abstract

1) the conclusion part of the abstract is not based on your finding, its a general kind of conclusion

Introduction

2)your in introduction is not well conceptualized, you did not show the gap for doing this research

you said "there is no paper locally" but there are number of papers on covid mitigation measures of health workers even in Ethiopia

methods

3) you have defined perceived susceptibility and risk of getting disease as the same variables when they actually are very different variable

4)you measured perceived severity and risk of getting severe disease as the same variable when they actually are different variable

5)you measured perceptions categorically which has a lot of limitation, what's your base to categorize perception?

i recommend you to treat perception items as continuous variable

6)how do you measured vaccine hesitancy, covid mitigation measures, perceived susceptibility and perceived severity please attach the questionnaire i want to see the questionnaire

RESULT PART

7)you have wrongly interpreted the odds ratios that are less than one

6. PLOS authors have the option to publish the peer review history of their article (what does this mean?). If published, this will include your full peer review and any attached files.

Reviewer #1: No

Reviewer #2: No

Reviewer #3: **Yes: **Abass Abdul-Karim

Reviewer #4: No

Reviewer #5: No

---

## [Author Response · Author response to Decision Letter 0]

26 Feb 2022

26/02/2022

Paavani Atluri, 

PLOS ONE

Dear Paavani Atluri, 

Subject: Submission of revised manuscript entitled as “Adherence of Covid-19 mitigation measures and its associated factors among health care workers at Referral Hospitals in Amhara Regional state of Ethiopia” (PONE-D-21-38230).

Thank you for email dated on January 21/2022 enclosing the Editorial member’s and the reviewer’s comments. We have carefully revised the manuscript and incorporated their comments accordingly. Our responses are given in point-by-point response below.

We hope the revised version is suitable for publication and look forward to hearing from you in due courses.

Sincerely

Beletech Fentie

University of Gondar, College of Medicine and health Sciences, School of Nursing, Department of pediatrics and child health nursing. 

Point by point responses to Editorial Board Member’s and Reviewers’ comments.

Title of paper: Adherence of Covid-19 mitigation measures and its associated factors among health care workers at Referral Hospitals in Amhara Regional state of Ethiopia

Editorial comments:

1. Please ensure that your manuscript meets PLOS ONE's style requirements, including those for file naming 

Authors Response: Thank you very much for your constructive comments and suggestions. We tried to incorporate your comments accordingly and we hope the manuscript meets PLOS ONE’s style requirements.

Authors’ Response: Thank you very much for your constructive comments. Written informed consent was obtained from each participant using communication channels (telegram and email) and those who agreed to participate were included in the study and this information is provided in the Ethics approval and consent to participate section of the manuscript. This study did not include the minors.

Authors’ Response: Thank you very much for your constructive comments. We tried to address the comments in the manuscript. Since the authors are in low income country to cover the cost, online grammar checker was used to correct the spelling and grammar errors (Grammarly.com)

4. We note that you have indicated that data from this study are available upon request. PLOS only allows data to be available upon request if there are legal or ethical restrictions on sharing data publicly.

Authors’ Response: Thank you very much for your comments. We have uploaded the minimal anonymized data set necessary to replicate our study findings as Supporting Information file.

5. PLOS requires an ORCID ID for the corresponding author in Editorial Manager on papers submitted after December 6th, 2016.

Authors’ Response: Thank you very much for your comment. The corresponding author has validated her ORCID ID in Editorial Manager.

Authors’ Response: Thank you very much for your comment. We deleted the ethics statement that was included other than the methods section in the manuscript.

7. Thank you for submitting the above manuscript to PLOS ONE. During our internal evaluation of the manuscript, we found significant text overlap between your submission and the following previously published works, some of which you are an author.

- https://journals.plos.org/plosone/article?id=10.1371%2Fjournal.pone.0257373

Authors’ Response: Thank you very much for your comments. We tried to revise the manuscript and rephrase the duplicated text and cite the sources. The published article you mentioned (https://journals.plos.org/plosone/article?id=10.1371%2Fjournal.pone.0257373) is conducted in the general community but the current study is conducted in health care providers which is different population in the general community(i.e The study populations of the already published article and the current manuscript is totally different). Therefore, we think there is no duplication.

8. Please review your reference list to ensure that it is complete and correct. If you have cited papers that have been retracted, please include the rationale for doing so in the manuscript text, or remove these references and replace them with relevant current references. Any changes to the reference list should be mentioned in the rebuttal letter that accompanies your revised manuscript. If you need to cite a retracted article, indicate the article’s retracted status in the References list and also include a citation and full reference for the retraction notice

Authors’ Response: Thank you very much for your comments. We have assessed the references as much we can and the references are correct and there is no retracted article cited.

Reviewer #1: 

1. Since your study is a form of survey study, could you add study map?? to have more description for the reader?

Author’s response: 

• Based on the comment, the authors incorporated a study map in the revised manuscript. 

2. Correct place for ethics approval and consent to participate subtitle, and no need to write duplicate subtitle.

Author’s response: 

• Thank you for the comments.

• Based on the comment, the authors removed the ethics section from the declaration part and only incorporated it in the method section. 

3. No need to describe the study area / objective under discussion again!! 

Author’s response: 

• Thank you. The authors removed it in the revised manuscript. 

4. Duplicate statements under abstract and conclusion.

Author’s response: 

• Thank you for your concern. In the revised manuscript, the authors rephrase the idea in the abstract section.

Reviewer #2: 

1. When you are writing the statement of the problem, it was nice if you put the paragraphs as follows; • Concise description of the problem (severity, which group is affected, the distribution of the problem, what contribute to the problem, the consequences of the problem. what policies, and strategies are in palace to combat the problem , what is known, what is not known, why you are interested in the topic ( what gaps exist).

Authors Response: Thank you very much for your constructive comments and suggestions. We tried to address your comments and suggestions accordingly in the manuscript.

2. Better to put in such way Binary logistic regression was be employed to identify factors associated with adherence of COVID-19 mitigation measures. Initially bivariate analysis was done and variables with p-value of below 0.2 were identified as candidate for multi-variable analysis. Then multi-variable analysis was done and adjusted odd ration was computed and interpreted. A p-value less than 0.05 is cut-off point for determining the significance of association. Result of the study was presented in text, table and graphs.

Author’s response: Based on the comments, the authors revised this paragraph under the subheading of the “Data processing and analysis” section of the manuscript. 

Reviewer #3:

 None

Author’s response: Reviewer 3 didn't have any comments to the authors regarding the manuscript.

Reviewer #4:

I appreciate the authors for doing a research on the current pandemic disease. But I have some concerns.

Author’s response:

•Thank you for your appreciation and positive feedback. 

1. Abstract is OK.

Author’s response: Thank you.

2. Methods: Please provide further detail how Random selection was carried out to select study participants.

Author’s response: Thank you for your concern. Based on the comments, the authors incorporated the details of randomization in the revised manuscript. 

3. Please try to provide the detail of the specific questions used to assess adherence level for all the three components.

Authors Response: Thank you very much for your constructive comments and suggestions. 

We tried to address all the issues raised in operational definition part of the revised manuscript.

4. Discussion: The justification provided by the authors on the discrepancy between the studies conducted in other part of the same country may be due to the difference in tool used to assess adherence. Please provide a clear justification why you preferred using a tool with only three components to assess adherence level when others used different tool (discussion part second paragraph line 13-14)

Authors Response: Thank you very much for your constructive comments.

The previous studies used more adherence components since they were conducted in the initial phase of the pandemic (like there were lockdown, no public transportation, satay at home rules, mass gathering and etc. in the world). Since stay at home, restriction of public transportation, mass gathering and any lockdown are removed; we used the three major components used to prevent the COVID- 19 pandemic. The three components are also highly recommended by world health organization and many other health authorities and organization including Ethiopian ministry of health. 

5. In discussion, the author only compared their finding with other studies on the adherence level and no comparisons were made with other studies for factors affecting adherence level.

Authors’ Response: Thank you very much for your constructive comments. In some extent, we tried to address the comment in the manuscript, but as our search we didn’t get any similar factors associated to adherence in other articles conducted in health care providers. That is why we left not discussed the factors variables. Instead, we tried to show scientific facts why these factors influence the adherence.

6. Discussion part should cover the interpretation of the finding, comparison with other studies, explanation for discrepancies if it exists, and the limitation of the study. The discussion part could be more than what the authors provided.

Authors Response: Thank you very much for your constructive comments. We tried to elaborate the discussion part as per your comment and suggestion

Reviewer#5

Dear authors, thank you for your effort, I have some comments and questions in your work

Author’s response: Thank you for your feedback.

Abstract:

1) The conclusion part of the abstract is not based on your finding, it’s a general kind of conclusion

Author’s response: Thank you. We revised and retyped it based on the findings of this study.

Introduction:

2) Your introduction is not well conceptualized, you did not show the gap for doing this research, you said "there is no paper locally" but there are number of papers on covid mitigation measures of health workers even in Ethiopia

Authors Response: Thank you very much for your constructive comment. We tried to address the comments in the manuscript. But still we couldn’t get any article published in the study setting among health care providers even in Ethiopia. 

Methods:

3) You have defined perceived susceptibility and risk of getting disease as the same variables when they actually are very different variable

Author’s response: 

According to the health belief model (HBM), 

• Perceived susceptibility is defined as a person’s subjective perception about their chance or risk of getting a certain condition, in this case, COVID-19.

• This means the literal definition of “Perceived susceptibility” is “perceived risk of getting a disease”.

• That was why the authors used the phrase “Perceived susceptibility/risk of getting COVID-infection”, which is to mean “perceived susceptibility of getting COVID-infection” or “perceived risk of getting COVID-infection”.

• In short, based on the definition of HBM, the authors used these two terms interchangeably. 

• But, to avoid ambiguity, the authors used only “perceived susceptibility” in the revised manuscript. 

4) You measured perceived severity and risk of getting severe disease as the same variable when they actually are different variable

Author’s response: 

• According to HBM, “perceived severity” refers to a person’s belief about the seriousness or severity of a disease. 

• i.e. Perceived severity of COVID-19 infection = Perceived risk of developing sever COVID-19 infection.

• That was why the authors used the term “perceived severity/risk of developing the severe disease”.

• But, to avoid ambiguity, the authors used only “perceived severity” in the revised manuscript. 

5) You measured perceptions categorically which has a lot of limitation, what's your base to categorize perception? I recommend you to treat perception items as continuous variable

Author’s response: 

• Thank you for your recommendation. But, we assessed it by using the Likert scale, as other literature has used before. After that, re-categorization was considered in a few variables if the chi-square assumption was not fulfilled.

6) How do you measured vaccine hesitancy, COVID-19 mitigation measures, perceived susceptibility and perceived severity please attach the questionnaire I want to see the questionnaire

Author’s response: 

• The way of measuring these variables was stated under the subheading “Operational Definition” of the method section. 

• Based on your request, we attached the questionnaire during the re-submission of the revised manuscript as a “supplementary file” . 

Result:

7) You have wrongly interpreted the odds ratios that are less than one

Author’s response:

We think the interpretations of the Odds rations are correct. For example:- AOR=0.15 means study participants adhered 15% to mitigation measures. In another way, it can be interpreted as study participants were 85% less likely adhere to mitigation measures of COVID-19. That is why we used one of the two ways of interpretations. If authors are mistaken, they can correct with reasonable justifications you will give us.

---

## [Decision Letter · Decision Letter 1]

22 Jul 2022

Adherence to Covid-19 mitigation measures and its associated factors among health care workers at Referral Hospitals in Amhara Regional State of Ethiopia

PONE-D-21-38230R1

Dear Dr. Fentie,

We’re pleased to inform you that your manuscript has been judged scientifically suitable for publication and will be formally accepted for publication once it meets all outstanding technical requirements.

Kind regards,

Paavani Atluri

Academic Editor

PLOS ONE

Additional Editor Comments (optional):

Reviewers' comments:

Reviewer's Responses to Questions

**Comments to the Author**

1. If the authors have adequately addressed your comments raised in a previous round of review and you feel that this manuscript is now acceptable for publication, you may indicate that here to bypass the “Comments to the Author” section, enter your conflict of interest statement in the “Confidential to Editor” section, and submit your "Accept" recommendation.

Reviewer #4: All comments have been addressed

2. Is the manuscript technically sound, and do the data support the conclusions?

Reviewer #4: Yes

3. Has the statistical analysis been performed appropriately and rigorously? 

Reviewer #4: Yes

4. Have the authors made all data underlying the findings in their manuscript fully available?

Reviewer #4: Yes

5. Is the manuscript presented in an intelligible fashion and written in standard English?

Reviewer #4: Yes

6. Review Comments to the Author

Reviewer #4: (No Response)

7. PLOS authors have the option to publish the peer review history of their article (what does this mean?). If published, this will include your full peer review and any attached files.

Reviewer #4: No

---

## [Editor Report · Acceptance letter]

28 Jul 2022

PONE-D-21-38230R1 

Adherence to Covid-19 mitigation measures and its associated factors among health care workers at Referral Hospitals in Amhara Regional State of Ethiopia 

Dear Dr. Fentie:

I'm pleased to inform you that your manuscript has been deemed suitable for publication in PLOS ONE. Congratulations! Your manuscript is now with our production department. 

Kind regards, 

on behalf of

Dr. Paavani Atluri 

Academic Editor

PLOS ONE